# Relationships between Strength, Ductility and Fracture Toughness in a 0.33C Steel after Quenching and Partitioning (Q&P) Treatment

Evgeniy Tkachev [1,2], Sergey Borisov [1,2], Yuliya Borisova [1,2], Tatiana Kniaziuk [1,3] and Rustam Kaibyshev [1,*]

1   Laboratory of Advanced Steels for Agricultural Machinery, Moscow Timiryazev Agricultural Academy, Russian State Agrarian University, 127550 Moscow, Russia; tkachev_e@bsu.edu.ru (E.T.); borisova_yu@bsu.edu.ru (Y.B.)
2   Laboratory of Mechanical Properties of Nanostructured Materials and Superalloys, Belgorod State National Research University, 308015 Belgorod, Russia
3   National Research Center "Kurchatov Institute"–Central Research Institute of Structural Materials "Prometey", 191015 St. Petersburg, Russia
*   Correspondence: kajbyshev@rgau-msha.ru; Tel.: +7-4722-585455; Fax: +7-4722-585417

**Abstract:** The effect of quenching and partitioning (Q&P) processing on strength, ductility and fracture toughness is considered in a 0.33% C-1.8% Si-1.44 Mn-0.58% Cr steel. The steel was fully austenitized at 900 °C and quenched to 210 °C for 30 s. Partitioning at 350 °C for 600 s produces a martensitic matrix with transition carbides, bainitic ferrite and film-like retained austenite (RA) that is stable against transformation to strain-induced martensite under tension. This processing provided the highest strength and fracture toughness but the lowest ductility and product of strength and elongation (PSE), $\sigma_B \cdot \delta$ (MPa·%). Partitioning at 500 °C produced RA with a relatively low carbon content and low volume fraction of carbides. The steel after this Q&P processing exhibits the highest ductility and PSE but low YS and Charpy V-notch (CVN) impact toughness. High ductility and PSE correlate with the ability of RA to transform into strain-induced martensite, while high strength and impact toughness are associated with the high-volume fraction of transition carbides in the carbon-depleted martensitic matrix and a lack of transformation of RA to strain-induced martensite. The highest CVN impact energy was attained in the steel exhibiting transgranular quasi-cleavage fracture with the lowest effective grain size for brittle fracture. No correlation between strength, ductility and fracture toughness is observed in Q&P steels if these materials have distinct structural constituents.

**Keywords:** quenching and partitioning; martensite; low-alloy steel; impact toughness; tensile properties; microstructure; phase transformation; carbides



## 1. Introduction

Low-alloy steels processed by quenching and partitioning (Q&P) are considered a third generation of advanced high-strength steel (AHSS) due to a unique combination of yield strength (YS), ultimate tensile strength (UTS), ductility and fracture toughness [1–3]. Q&P steels are subjected to complex heat treatment involving austenitizing followed by rapid cooling to a temperature, $T_q$, between the martensite-start ($M_S$) and martensite-finish ($M_F$) temperatures [1–3]. This step of the Q&P process produces martensite and a considerable amount of retained austenite (RA) [1–3]. The second step is carried out at a temperature, $T_P$, to provide the carbon partitioning from the martensite to retained austenite (RA). Usually, the temperature of partitioning, $T_P$, is higher than $M_s$, and, therefore, $T_P > T_q$ [1–3]. Two accompanying processes usually occur during Q&P processing. The first process is the precipitation of the η(Fe₂C)- or ε(Fe₂.₄C)- transition carbides in a martensitic matrix that leads to additional carbon depletion from supersaturated martensite [2–5]. This low-temperature tempered martensite is often termed primary martensite [2]. The second process is the decomposition of untransformed austenite to bainitic ferrite. Q&P steels are

usually alloyed with $\geq 1.5$ wt.% Si, which suppresses the formation of cementite during Q&P processing if the $T_p$ and $T_q$ temperatures are below 400 °C [3,4,6,7]. The partitioning step is followed by air cooling or water quenching, which may result in the formation of secondary untempered martensite with a higher carbon content compared to the primary martensite [2,4,5,8,9]. Thus, the Q&P process produces a multiphase structure consisting of carbon-depleted martensite with transition carbides, carbon-enriched martensite, carbon-depleted bainitic ferrite and carbon-enriched RA [1–5,8–10]. For low- and medium-carbon steels, both the primary and the secondary martensite usually hold the Kurdjumov–Sachs (K-S) orientation relationship with the austenite matrix, which can be determined by electron backscatter diffraction (EBSD) analysis [5,11–13].

Two types of RA distinguished by morphology and carbon content may evolve in Q&P steels [2,3,5,8,11,14]. Blocky-type RA is usually located in the triple junctions of prior austenite grains (PAGs) and martensite packets [4,11,15]. Q&P steels with blocky-type RA have a carbon content of above 0.3 wt.% and exhibit elongation-to-failure, $\delta$, $\geq 15\%$ with a product of strength and elongation (PSE), $UTS \times \delta$ (MPa $\times$ %), ranging from $2 \times 10^4$ to $4 \times 10^4$ MPa·% due to transformation of blocky-type RA to strain-induced martensite under tension [1,8,11,16–18]. The relatively high YS of above 1 GPa of these Q&P steels is attributed to the combination of the solid solution strengthening of blocky-type RA associated with carbon enrichment of this phase of $C \geq 1.14$ wt.% and dislocation strengthening associated with the high dislocation density in RA [11]. The high values of the PSE correlate with the high absorbed impact energy ranging from 30 to 110 J [8,14,19–21]. These data support the general statement that a high PSE indicates a high fracture resistance [1]. Additionally, the correlation between the PSE and fracture toughness for automotive steel sheets was recently argued [22,23]. Conversely, no clear relationship between fracture toughness and the PSE was found in a 0.3C-2.5Mn-1.5Si-0.8Cr steel processed by Q&P [23]. Chong Gao et al. [22] assume that there exist some exceptions from the basic rule that a high PSE means better fracture toughness since crack initiation in Q&P steels is mainly controlled by the parameters of the martensitic phase, while ductility is controlled by the volume fraction, morphology of RA and its susceptibility to transformation to strain-induced martensite under tension [11,23,24].

The lack of the proportional dependence between the PSE and CVN impact energy was observed in Q&P steels with $C \leq 0.3$ wt.% that exhibit PSE values ranging from $1.0 \times 10^4$ to $2.7 \times 10^4$ MPa·% with YS ranging from 1050 to 1250 MPa and CVN impact energy ranging from 60 to 110 J [21,25–28]. Film-like RA dominates in these Q&P steels [5,27–29]. Kai Yang et al. [28] showed that the combination of the ductility, PSE and Charpy V-notch (CVN) impact absorbed energy mainly depends on the stability of the film-like RA during tensile or impact tests [30]. S. Takaki et al. [30] showed that film-like RA is characterized by increased stability to transformation to strain-induced martensite compared to blocky-type RA. The relationship between the PSE and fracture toughness depends on RA morphology, volume fraction and stability as well as characteristics of the martensitic matrix and other constituent phases [24]. Unfortunately, the effect of the precipitation of carbide particles and characteristics of RA on the combinations of strength, ductility and impact toughness is scarcely reported for Q&P steels.

The aim of the present work is to examine the relationships between strength, ductility, PSE and CVN impact absorbed energy in a low-alloy steel with 0.33 wt.% C processed by Q&P. The effects of the tempering of the primary martensite and the stability of RA on the mechanical properties are discussed. For this purpose, three different Q&P routes were followed to obtain microstructures containing various volume fractions of bainitic ferrite and secondary martensite, as well as the different morphology and carbon content of RA. This study is a part of a complex work and data on the structure and mechanical properties of this steel subjected to quenching and tempering (Q&T) processing are available [4] for comparison.

## 2. Material and Methods

The chemical composition of the studied low-alloy medium-carbon steel is presented in Table 1.

**Table 1.** Chemical composition of the steel in wt.%.

| Fe | C | Si | Mn | Cr | N | P | S |
|---|---|---|---|---|---|---|---|
| Balance | 0.33 | 1.85 | 1.44 | 0.58 | 0.0084 | $\leq 0.01$ | $\leq 0.007$ |

The steel was produced by air induction melting followed by electro-slag remelting [4]. Then the solution treatment was performed at 1150 °C for 4 h followed by hot forging at temperatures of 1150–950 °C into billets with dimensions of $60 \times 150 \times 450$ mm$^3$. Samples with 12 mm thickness were machined from the billets and austenitized at 900 °C for 10 min followed by quenching in a salt bath at $T_q$ = 210 °C for 30 s (Figure 1). Next, the samples were subjected to isothermal holding at $T_p$ = 350 °C for 600 s and at $T_p$ = 500 °C for 20 s and 100 s in a second salt bath followed by water quenching (WQ) (Figure 1).

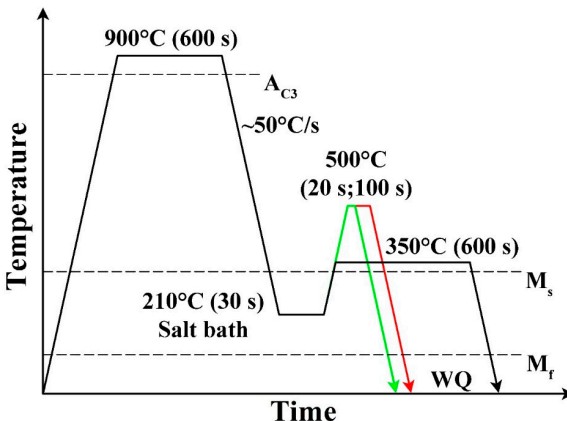

**Figure 1.** Schematic diagram of Q&P routes.

The Rockwell hardness was measured using a Wolpert Wilson hardness tester 600 MRD (Illinois ToolWorks Inc., Norwood, MA, USA) following the ASTM E18 standard. Ten indentations were performed for each sample. Uniaxial tension tests were carried out using an Instron 5882 testing machine (Illinois ToolWorks Inc., Norwood, MA, USA) at a loading rate of 1 mm/min and ambient temperature in accordance with the ASTM E08M-04 standard. Two tensile specimens with a gauge length of 35 mm and a cross-section of 7 mm $\times$ 3 mm were tested for each material condition. The elongation-to-failure was measured by the non-contact digital image correlation (DIC) method. CVN specimens with dimensions of $10 \times 10 \times 55$ mm$^3$ and a 2 mm V-notch were tested using an Instron IMP460 impact testing machine (Illinois ToolWorks Inc., Norwood, MA, USA) equipped with an Instron Dynatup Impulse data acquisition system in accordance with the ASTM E-23 standard.

Structural characterization was carried out by scanning electron microscopy (SEM) and transmission electron microscopy (TEM). Samples for SEM and thin foils for TEM observations were prepared by electrolytic polishing and the twin-jet polishing technique, respectively, using 10% $HClO_4$ and 90% $CH_3COOH$ solution. A Quanta 600 FEG (FEI Corporation, Hillsboro, OR, USA) SEM operating at 20 kV and equipped with an electron backscatter diffraction (EBSD) pattern analyzer and EDAX Velocity™ camera was used for orientation imaging microscopy (OIM) and observations of etched surfaces and fracture surfaces. The OIM images were subjected to a clean-up procedure, setting the minimal confidence index to 0.1. The high-angle boundaries (HAB) were defined with a misorientation $\geq 15°$ and depicted on the inverse pole figure maps as black lines. Dislocation density characterizing the elastic bending of the crystal lattice was calculated from EBSD maps

using the Kernel misorientation function with the presumption that all dislocations have the same Burgers vector as [31]:

$$\rho_{KAM} = \frac{2 \cdot \theta_{KAM}}{b \cdot h},$$ (1)

where $b$ is the Burgers vector and $h$ is the scanning step size. Reconstructruction of PAGs and martensite packets was carried out using the method proposed by Niessen et al. [32] and assuming the formation of a lath martensite structure in accordance with the K-S orientation relationship (OR) between the austenite, $\gamma$, and martensite, $\alpha'$ [15]:

$$\{111\}_\gamma \| \{011\}_\alpha, \langle \overline{1}01 \rangle_\gamma \| \langle \overline{1}\overline{1}1 \rangle_\alpha$$ (2)

A Jeol JEM-2100 (Jeol Ltd., Tokyo, Japan) operating at 200 kV and equipped with an INCA energy dispersive X-ray spectrometer (EDS) was used for TEM studies.

Volume fraction of RA was determined by the magnetic saturation method using a Fischer Feritscope FMP30 (Helmut Fischer Gmbh, Sindelfingen, Germany) and X-ray analysis using a Rigaku Ultima IV diffractometer (Rigaku Co., Tokyo, Japan) using Cu K$_\alpha$ radiation with a step size of 0.02 deg, at 40 kV and 40 mA. The austenite $(111)_\gamma$ and $(200)_\gamma$ diffraction peaks and the martensite $(110)_{\alpha'}$, $(200)_{\alpha'}$ and $(220)_{\alpha'}$ diffraction peaks were analyzed by comparing their integrated intensities [33].

The carbon content in RA was calculated using the following equation [11,34,35]:

$$a_\gamma(A) = 3.578 + 0.033C_{RA} + 0.00095Mn_{RA} + 0.0031Mo_{RA}$$ (3)

where $C_{RA}$, $Mn_{RA}$ and $Mo_{RA}$ are the concentrations of the three solutes in RA in wt.%. Carbon concentration was derived from X-ray data and Mn and Mo concentrations were taken as content of these elements in the steel since no substitutional atom partitioning occurs under Q&P [3,4]. The effect of Si on the lattice parameter of RA was discarded since the influence of Si on $a_\gamma$ values is negligible [34,36].

Dilatation experiments were performed using a Bähr DIL 805 A/D dilatometer (TA Instruments Inc., New Castle, DE, USA) on cylindrical samples with a 10 mm length and a 3 mm diameter. Other details of experimental techniques were reported in previous works [4,5,11,37].

## 3. Results

### 3.1. Phase Transformations

Phase transformations during Q&T processing are considered in the previous work [4] in detail. The $A_{C3}$ and $A_{C1}$ critical temperatures are 758 and 817 °C, respectively, [4] and, therefore, austenitizing at 900 °C provides a fully austenitic structure. The $M_s$ and $M_f$ temperatures were determined by dilatometry as 325 and 104 °C, respectively, and the volume fraction of primary martensite at the selected quenching temperature of 210 °C is 0.78 (Figure 2a). Several empirical relationships were developed to estimate the $M_s$ from chemical composition [38–40]. Two equations suggested by K. W. Andrews [40] and C. Capdevila et al. [41] are used for the prediction of $M_s$ temperatures in the low-alloy steels:

$$M_s(°C) = 539 − 423\% \text{ C} − 30.4\% \text{ Mn} − 12.1\% \text{ Cr} − 7.5 \text{ Si}$$ (4)

$$M_s(K) = 764.2 − 302.6\% \text{ C} − 30.6\% \text{ Mn} − 8.9\% \text{ Cr} − 14.5\% \text{ Si}$$ (5)

where the concentrations of carbon, manganese, chromium and silicon are in wt.%. The $M_s$ temperatures calculated from Equations (4) and (5) are $M_s(°C) = 334$ °C and $M_s(K) = 588$ K (315 °C), respectively. Therefore, the two empirical formulas predict $M_s$ with high accuracy in the present steel and could be used to calculate the $M_s$ for carbon-enriched RA. It is worth noting that Formula (4) overestimates the $M_s$ by $\Delta T = 9$ K, and Formula (5) underestimates the $M_s$ by $\Delta T = 10$ K. The η-Fe$_2$C transition carbides precipitate in the martensitic matrix at

temperatures of 200–400 °C and cementite precipitates along HABs and LABs at T ≥ 474 °C during conventional tempering [4]. The decomposition of RA to bainitic ferrite and the transition carbides occurs at tempering temperatures ranging from 200 to 400 °C [4]. This agrees well with the dilatometric data obtained during Q&P processing (Figure 2b,c). The observed length increase at the partitioning stage at 350 °C can be attributed to the bainitic transformation, whereas the length contraction during partitioning at 500 °C is associated with the carbide precipitation inside austenite [42]. Based on the dilatometric data and X-ray analysis, the fractions of bainitic ferrite, secondary martensite, ferrite + carbides and RA were calculated. The obtained volume fractions of the microstructure constituents are summarized in Table 2. According to these results, the increase in the partitioning time at 500 °C leads to a noticeable increase in the volume fraction of the ferrite + carbides and decrease in the volume fraction of secondary martensite. The precipitation of cementite at this temperature is accompanied by the dissolution of the transition carbides, whereas no overlap between the precipitation of these two types of carbides was observed at tempering temperatures below 400 °C [4].

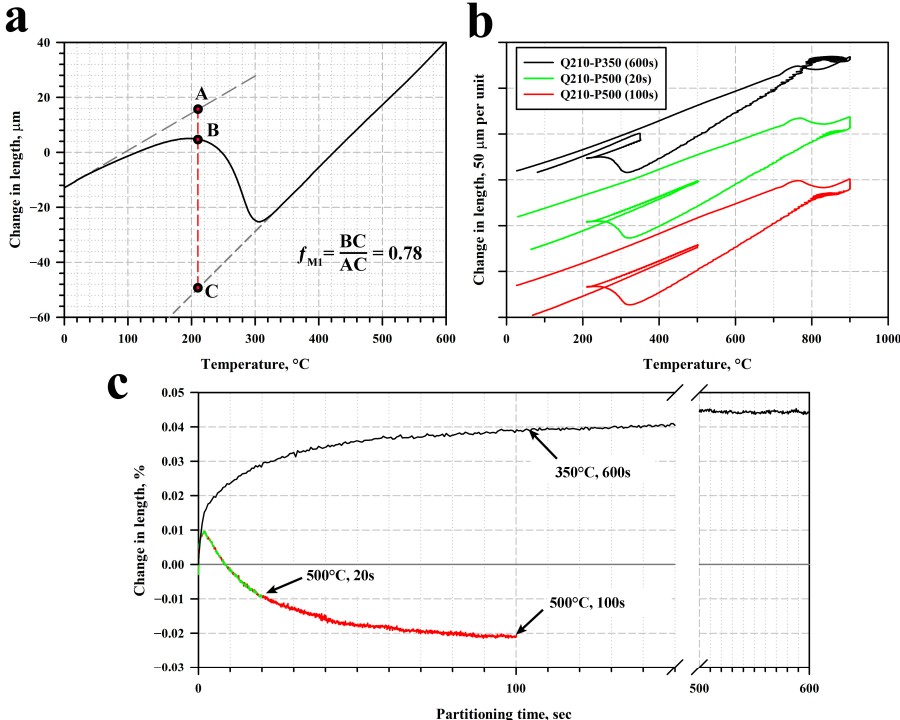

**Figure 2.** Change in length versus temperature obtained during water quenching of the steel (**a**) and during Q&P processing (**b**). Relative change in length as a function of partitioning time during Q&P processing (**c**).

**Table 2.** Volume fractions of the microstructure constituents after Q&P processing.

| Q&P Processing | Primary Martensite | RA | Bainitic Ferrite | Secondary Martensite | Ferrite + Carbides | M$_S$ for Secondary Martensite | Carbon Content in Secondary Martensite (Equation (4)) |
|---|---|---|---|---|---|---|---|
| Q210-P350 (600 s) | | 0.10 | 0.07 | 0.05 | - | 201 °C | 0.62 wt.% |
| Q210-P500 (20 s) | 0.78 | 0.11 | - | 0.08 | 0.03 | 253 °C | 0.50 wt.% |
| Q210-P500 (100 s) | | 0.08 | - | 0.04 | 0.10 | 174 °C | 0.68 wt.% |

The selected T$_q$ temperature is significantly above the martensite-finish temperature that provides a high volume fraction of untransformed austenite [2,3]. Volume fractions of martensite, $f_{\alpha'}$, and untransformed austenite prior to partitioning ($f_\gamma^{in} = 1 - f_{\alpha'}$) could be evaluated using the Koistinen–Marburger (K-M) equation [2,9,43]:

$$f_{a\prime} = 1 - \exp\left(-1.1 \cdot 10^{-2}\left(M_s - T_q\right)\right), \tag{6}$$

The calculated $f_{\alpha\prime}$ and $f_\gamma^{in}$ values are 72% and 28%, respectively.

### 3.2. Microstructure after Q&P Processing

Typical SEM microstructures are shown in Figure 3. Primary martensite is heavily etched [11,26,44,45] and contains carbide particles readily distinguished by secondary electron imaging, which is indicative of a carbon-depleted structural component. Areas of secondary martensite and RA are distinctly revealed by slight etching due to the carbon enrichment. The secondary martensite and RA are hardly distinguished in these areas, which is indicative of the similar carbon content in these structural constituents (Figure 3d). No carbide precipitation could be observed in carbon-enriched areas after partitioning at a temperature of 350 °C (Figure 3a), while some carbides are present in the carbon-enriched matrix after partitioning at 500 °C for 20 s (Figure 3b). The formation of carbide chains along block boundaries and the boundaries between the primary martensite and carbon-enriched areas is observed after partitioning at 500 °C for 100 s (Figure 3c).

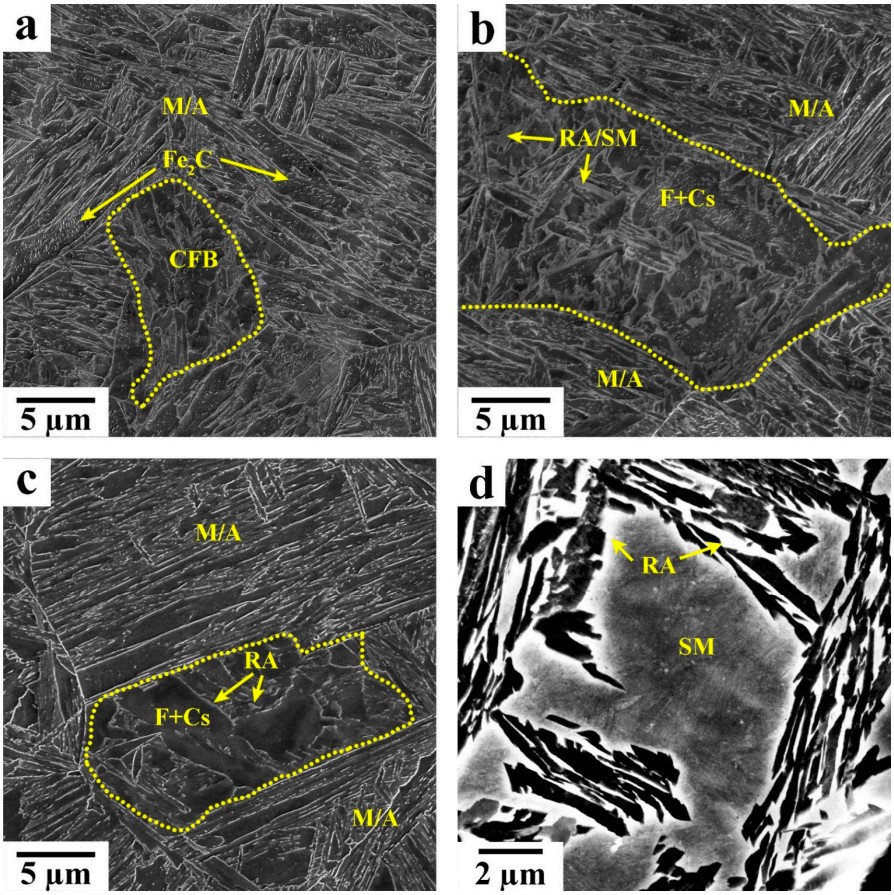

**Figure 3.** SEM microstructures of Q&P samples: (**a**) Q210-P350 (600 s), (**b**) Q210-P500 (20 s) and (**c**) Q210-P500 (600 s). (**d**) Coarse RA/SM island in the Q210-P500 (20 s) sample. M/A is the martensite/austenite constituent, CFB is the carbide-free bainite, SM is the secondary martensite and F + Cs is the ferrite + carbides.

Q&P processing with partitioning at 350 °C leads to the subdivision of PAGs and packets of primary lath martensite [15] to rectangular areas with a thickness of ~1.5 μm by carbon-enriched areas (Figure 3a). Unitized packets of primary martensite with an average size of 5 μm and irregular shape are rarely observed (Figure 3a). After partitioning at 500 °C for 20 s, this subdivision remains (Figure 3b). After partitioning at 500 °C for

100 s, the packets of primary martensite are largely subdivided into isolated areas by the well-defined HABs with chains of boundary carbides (Figure 3c).

Typical misorientation maps are presented in Figure 4 and structural parameters are summarized in Table 3. The lath martensite exhibits a three-level hierarchy in its morphology, i.e., blocks, packets and PAG [15]. After partitioning at 350 °C, the ratio between the average dimensions of martensite packets and PAGs is $D_{packet}$ ~0.16$D_{PAG}$ and the ratio between the block width and the average size of packets is $d_{block}$ ~0.25$D_{packet}$ (Table 3). However, the distribution of PAG dimensions is non-uniform. Coarse PAGs with dimensions of above 30 μm alternate with fine PAGs with sizes below 20 μm. The EBSD analysis suggests that the small martensite packets of secondary martensite are often located in the coarse primary martensite and surrounded by RA (Figure 4). Coarse packets with an irregular shape are often observed in the PAGs (Figure 4(a2,a3)). These packets consist of ≥4 blocks with different K-S OR, whereas the small packets consist of two blocks or even one block. As a result, the average distance between HABs is 70% larger than the block thickness (Table 3).

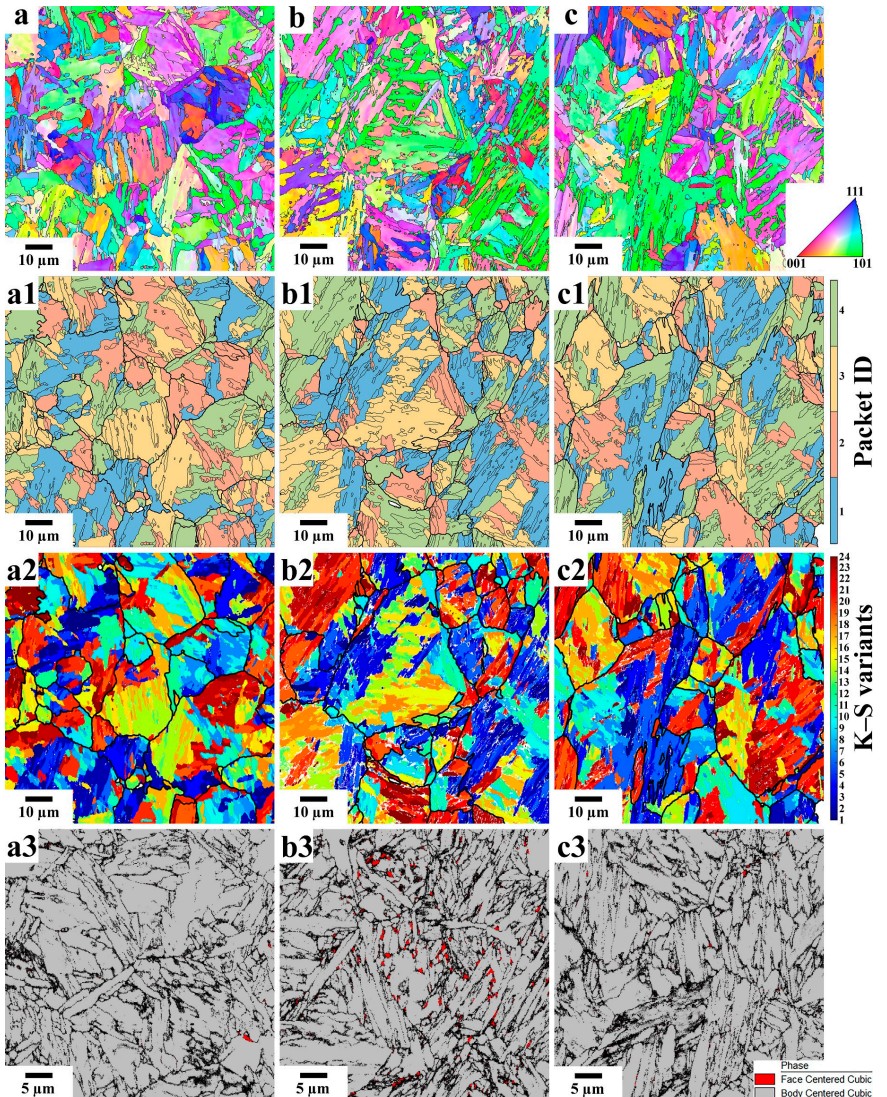

**Figure 4.** EBSD analysis of the steel after partitioning at 350 °C for 600 s (**a–a3**), at 500 °C for 20 s (**b–b3**) and at 500 °C for 100 s (**c–c3**). Inverse pole figure maps (**a–c**), EBSD maps showing the identification numbers of martensite packets and the reconstructed boundaries of PAGs (thick black lines) and HABs (thin black lines); (**a1–c1**), K–S variants maps (**a2–c2**) and phase maps (**a3–c3**).

**Table 3.** Effect of Q&P regime on the average size of PAGs, $d_{PAG}$, packets, $d_P$, spacing between HABs, $d_{HAB}$, thickness of laths, $d_l$, and dislocation densities calculated from EBSD data, $\rho_{KAM}$.

| Q&P Processing | $d_{PAG}$ (SEM), μm | $d_P$ (EBSD), μm | $d_b$ (EBSD), μm | $d_{HAB}$ (EBSD), μm | $d_l$ (TEM), nm | $\rho_{KAM}$, $m^{-2} \times 10^{14}$ |
|---|---|---|---|---|---|---|
| Q210-P350 (600 s) |  | 4.1 ± 0.5 | 1.0 ± 0.2 | 1.7 ± 0.2 | 220 ± 15 | 5.8 |
| Q210-P500 (20 s) | 26.4 ± 3.3 | 4.7 ± 0.7 | 0.9 ± 0.2 | 1.4 ± 0.2 | 186 ± 15 | 6.1 |
| Q210-P500 (100 s) |  | 4.1 ± 0.5 | 1.0 ± 0.2 | 1.5 ± 0.2 | 262 ± 17 | 6.0 |

Galindo-Nava et al. assumed [46] that the relationships between the size of PAGs, packets and blocks can be predicted using the following equations:

$$D_{packet} = \sqrt{\frac{3\sqrt{3}}{8N_p}D_{PAG}} \qquad (7)$$

$$d_{block} = \frac{1}{N_b}D_{Packet} \qquad (8)$$

where $N_p$ is the number of packets in a PAG, and $N_b$ is the number of blocks in a packet, which provides the accommodation of lattice distortions associated with $\gamma \rightarrow \alpha'$ transformation. Equation (8) is fulfilled for $N_b = 4$, which agrees with experimental observations. However, four blocks per packet are rarely observed. Thus, Equation (7) could not be used to describe the structural parameters of the lath martensite structure. The microstructures after Q&P processing are characterized by a high degree of distortion associated with $\rho_{KAM}$ values (Table 3). This can be associated with long-range stress fields from the lath boundaries of primary and secondary martensite (Table 3).

Three types of other phases are distinguished in the lath martensite structure after Q&P. Film-like RA is revealed by TEM observations (Figure 5) and blocky-type RA could be observed by the EBSD technique (Figure 4(b3)). The parameters of RA are summarized in Table 4. The volume fraction of the blocky-type RA in the microstructure after partitioning at 350 °C for 600 s and after partitioning at 500 °C for 100 s is negligible. After partitioning at 350 °C, the films of RA are relatively thick and the carbon content in this phase is high (Table 4). Partitioning at 500 °C decreases the carbon content in RA and increasing the partitioning time from 20 s to 100 s decreases the average thickness of the RA films (Table 4). The $M_s$ values of RA were calculated with the presumption of a paraequilibrium condition existing, i.e., no partitioning of substitutional solutes takes place between the martensitic/ferritic matrix and RA during Q&P processing [3,47]. It can be seen (Table 4) that the increasing carbon content in the austenite increases strongly the difference in values calculated using the equations suggested by K. W. Andrews [40] and C. Capdevila et al. [41] (Table 4). Partitioning at 350 °C decreases the $M_s$ down to the ambient temperature or below (Table 4) and, therefore, the transformation of RA to the martensite could not occur during the final cooling to room temperature. After partitioning at 500 °C, the feasibility of a martensitic transformation occurring during cooling is ambiguous (Table 4). It is apparent that the transformation of RA to bainite may play an important role in decreasing the volume fraction of RA during partitioning. The bainite-start temperature calculated by the empirical formula

$$B_s(°C) = 839 - 86\% \text{ Mn} - 67\% \text{ Cr} + 23\% \text{ Si} - 271 \cdot (1 - \exp(-1.33\% \text{ C})) \qquad (9)$$

showed excellent agreement with the experimental data for steels with a carbon content below 1 wt.% and Si additions [48].

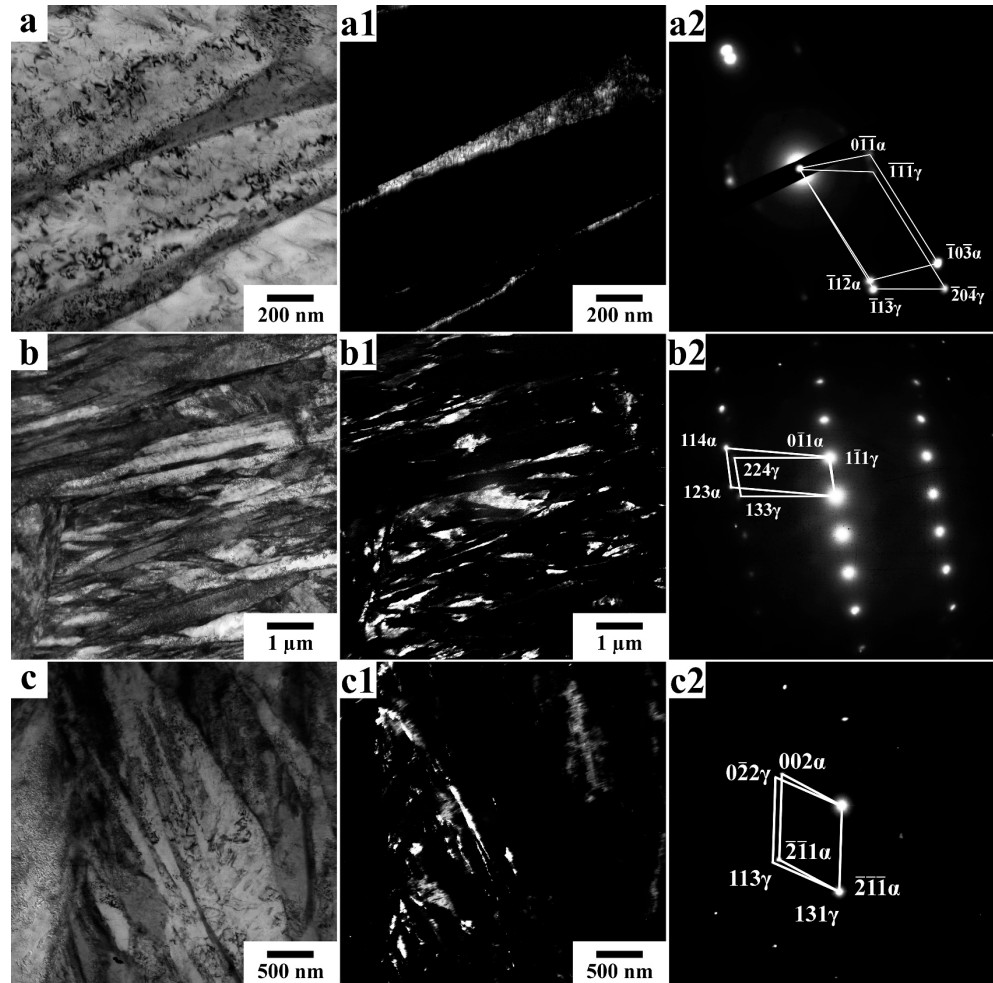

**Figure 5.** Bright-field images (**a**–**c**) and dark-field images (**a1**–**c1**) of RA and selected area electron diffraction pattern (SAED) (**a2**–**c2**) after quenching and partitioning at 350 °C for 600 s (**a**–**a2**), 500 °C for 20 s (**b**–**b2**) and 500 °C for 100 s (**c**–**c2**).

It can be seen (Table 4) that the bainitic transformation could occur during partitioning at 350 °C. However, the bainitic transformation could not contribute significantly to a decrease in the volume fraction of RA at a temperature of 500 °C since the $B_S$ and partitioning temperatures are nearly the same (Table 4). Therefore, the martensitic transformation is the most likely mechanism providing a decrease in the volume fraction of RA after partitioning at 500 °C.

**Table 4.** Effect of Q&P processing on characteristics of RA.

| Temperature and Duration of Partitioning Stage | Thickness of Film-Like RA, nm | Volume Fraction of RA [1], % | Carbon Content in RA (X-ray), wt.% | $M_S$ (Dilatometry) [2], °C | $B_S$ (Equation (9)), °C | Volume Fraction of RA after Tension (X-ray) [1], % |
|---|---|---|---|---|---|---|
| 350 °C for 600 s | 125 ± 10 | 10/10 | 1.3 | −74/21 | 495 | 9/7 |
| 500 °C for 20 s | 175 ± 15 | 11/10 | 1.0 | 53/112 | 519 | 6/4 |
| 500 °C for 100 s | 80 ± 10 | 8/5 | 1.1 | 11/82 | 509 | 2/2 |

[1] Numerator and denominator are volume fractions obtained by X-ray and magnetic saturation measurement techniques, respectively. [2] Numerator and denominator are martensite-start temperatures calculated by Equations (4) and (5), respectively.

The second phase is transition η-Fe$_2$C carbide particles precipitatated during Q&P processing (Figure 6, Table 5). These carbides exhibit plate-like shape with a high diameter-to-thickness aspect ratio (AR) of ~9, which is indicative of diffusion-controlled lengthening [49]. Transition carbides are observed after all Q&P processing conditions. The third

phase is cementite particles, which were detected after partitioning at 500 °C for 100 s (Figure 6c). The number density of cementite is insignificant and only separate particles are observed, which is in contrast to conventional tempering at this temperature [4]. However, precipitation of cementite leads to almost full dissolution of the transition carbides. Partitioning at 500 °C with a duration of up to 100 s could be considered rapid tempering since no extensive precipitation of cementite occurs [50,51]. It should be noted that cementite particles were not observed in the microstructure after partitioning at 500 °C for 20 s (Figure 6b).

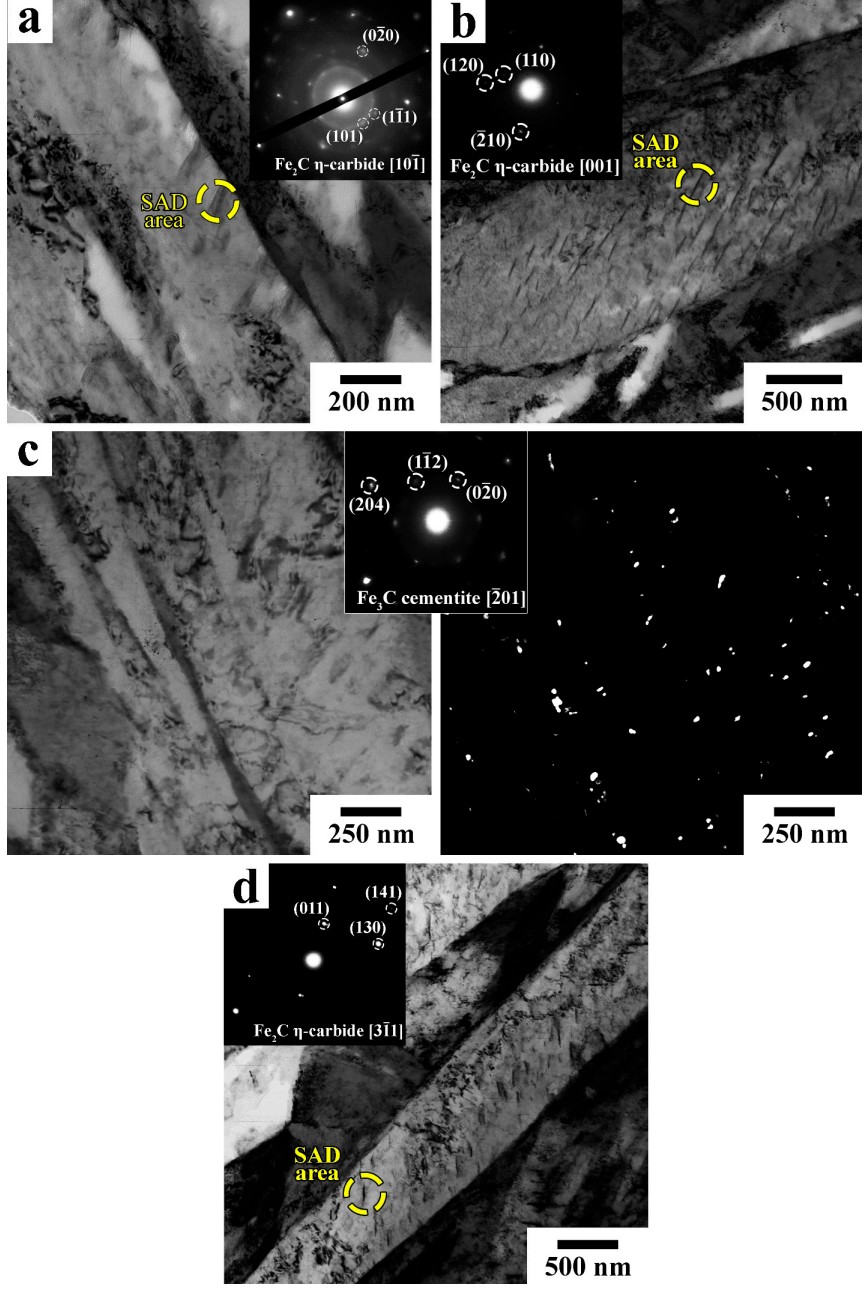

**Figure 6.** Typical TEM micrographs after Q&P processing (**a**) partitioning at 350 °C for 600 s, bright-field image and SAED from transition η-Fe$_2$C carbide; (**b**) partitioning at 500 °C for 20 s; (**c**) partitioning at 500 °C for 100 s, dark-field image in cementite spot with SAED; (**d**) partitioning at 500 °C for 100 s, bright-field image and SAED from η-Fe$_2$C carbide.

**Table 5.** Effect of tempering on the dimensions of carbides.

| Temperature and Duration of Partitioning Stage | Dimensions of Transition Carbides *, nm | Average Size of Cementite, nm |
|---|---|---|
| 350 °C for 600 s | 90/11 | - |
| 500 °C for 20 s | 126/13 | - |
| 500 °C for 100 s | 95/11 | 19 |

\* Numerator and denominator are the particle length and thickness, respectively.

### 3.3. Effect of Q&P Processing on Mechanical Properties

The engineering tensile stress–strain curves of the steel after Q&P processing are shown in Figure 7a and the hardness, YS, UTS, ductility and PSE are summarized in Table 6. The steel exhibits continuous yielding [38] followed by work hardening in all Q&P conditions. After partitioning at 350 °C, the strain hardening stage is short, which leads to relatively small uniform elongation. It is worth noting that the uniform elongation after this processing (Table 6) and Q&T treatment with tempering at 280 and 400 °C [4] is nearly the same. Increasing the partitioning temperature and duration of the partitioning stage decreases YS and increases strain hardening rate, $d\sigma/d\varepsilon$, (Figure 7b) at a true strain of above 0.02. The onset of the Considère condition [38]

$$\frac{d\sigma}{d\varepsilon} = \sigma \tag{10}$$

has no relation to the transition from stable plastic flow to unstable plastic flow for the steel in this condition (Figure 7a,b). However, the onset of necking is observed at relatively high strain (Figure 7a), which provides a ~50% increase in ductility and the PSE in comparison with conventional tempering at 280 and 400 °C [4]. Thus, the Q&P processing increases ductility at the expense of YS, UTS and hardness (Table 6) in comparison with Q&T treatment if the partitioning and tempering temperatures are close [4].

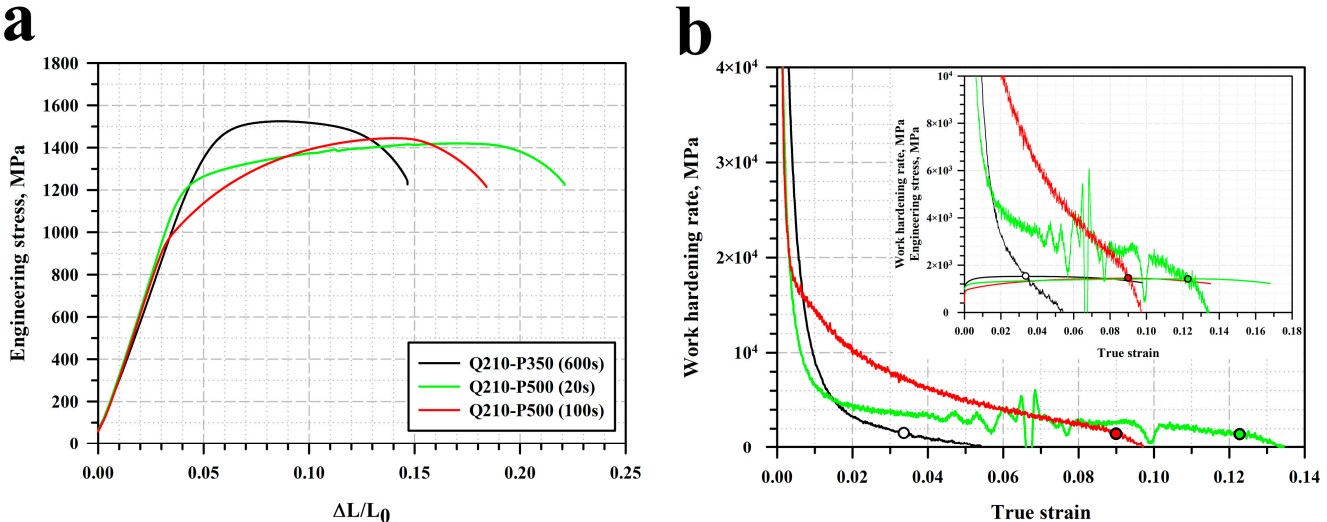

**Figure 7.** Engineering stress–strain curves (**a**) and strain hardening vs. strain curves (**b**).

**Table 6.** Effect of Q&P processing on mechanical properties.

| Temperature and Duration of Partitioning Stage | YS, MPa | UTS, MPa | Elongation-to-Failure, % | PSE, GPa·% | Uniform Elongation, % | HRC |
|---|---|---|---|---|---|---|
| 350 °C for 600 s | 1310 ± 10 | 1530 ± 10 | 10 ± 1 | 15.6 | 3.4 | 46 |
| 500 °C for 20 s | 1180 ± 10 | 1430 ± 10 | 19 ± 2 | 25.8 | 12.0 | 43 |
| 500 °C for 100 s | 950 ± 20 | 1450 ± 10 | 14 ± 1 | 20.8 | 9.3 | 42 |

At $T_p$ = 500 °C, the onset of the Considère condition shifts to higher strain and correlates with a transition to unstable plastic flow. This leads to an increase in uniform elongation by a factor of 3 but decreases YS with increasing temperature and duration of the partitioning stage (Table 6). UTS value and hardness decrease insignificantly. The ~30% and ~60% increase in the PSE takes place after partitioning at 500 °C for 100 s and 20 s, respectively (Table 6).

The highest CVN impact energy is observed in the steel sample after partitioning at 350 °C (Table 7). Q&P processing increases the impact energy by a factor of ~3 in comparison with Q&T treatment if the partitioning and tempering temperatures are close [4]. However, partitioning at 500 °C decreases the impact energy significantly (Table 7). After partitioning at 500 °C for 20 s, the value of CVN impact energy is two times smaller than that after tempering at 500 °C [4]. It is worth noting that after partitioning at 500 °C for 20 s, the steel exhibits the highest PSE and lowest CVN impact energy, whereas the lowest PSE and highest adsorbed energy were observed after partitioning at 350 °C (Tables 6 and 7). Therefore, no correlation between these two characteristics of fracture resistance takes place in Q&P steels. Increasing the duration of the partitioning stage at 500 °C increases the CVN impact energy and decreases the PSE (Tables 6 and 7).

**Table 7.** The CVN impact energy, dynamic yield strength (DYS), dynamic ultimate tensile strength (DUTS), the maximum load, $P_M$, ductile fracture initiation energy, $E_i$, fracture propagation energy, $E_p$, and total fracture energy, E, calculated from CVN load–deflection curves.

| Temperature and Duration of Partitioning Stage | CVN, J | $P_{GY}$, kN | $P_M$, kN | $P_A$, kN | DYS, MPa | DUTS, MPa | $E_i$, J | $E_p$, J | E, J |
|---|---|---|---|---|---|---|---|---|---|
| 350 °C for 600 s | 58 | 32 | 40 | 2.7 | 1400 | 1570 | 37 | 21 | 58 |
| 500 °C for 20 s | 16 | 21 | 23 | - | 915 | 915 | 9 | 7 | 16 |
| 500 °C for 100 s | 38 | 25 | 32 | 2.5 | 1110 | 1270 | 24 | 14 | 38 |

Two types of load–deflection curves can be distinguished in Figure 8. First, after partitioning at 350 °C for 600 s and 500 °C for 100 s, the curves exhibit the general yield load, $P_{GY}$, associated with the onset of ductile crack growth, the maximum load, $P_M$, and the $P_A$ point associated with the final crack arrest before the formation of shear lips [52–55]. The $P_M$ values are significantly higher than the $P_{GY}$ values and the difference in the deflection between these two points is about 0.7 mm. Therefore, ductile crack extension occurs at a relatively large displacement between the $P_{GY}$ and $P_{IF}$ points [52,55] and the highest portion of the impact energy is adsorbed due to the initiation of a crack with critical dimension, $E_i$, (Table 7) followed by crack propagation. It is worth noting that at the crack propagation stage, the high energy, $E_p$, is adsorbed (Table 7), and, therefore, no unstable crack propagation takes place. This behavior can be interpreted in terms of fast crack propagation and no characteristic point of the onset of unstable fracture, $P_F$, [4,52–57] could be distinguished. The arrest of crack propagation appears at very small loads and, therefore, insignificant energy is adsorbed at this stage.

The second type of load–displacement curve is observed after partitioning at 500 °C for 20 s (Figure 8). The general yield load is poorly distinguished from the $P_M$ point and the highest energy is adsorbed at the crack propagation stage. The $E_i$ value is relatively small (Table 7). No crack arrest stage is observed (Figure 8). It is worth noting that the steel processed by Q&P exhibits a specific dynamic impact behavior that is distinctly distinguished from that observed in the low-alloy high-strength steels after Q&T treatment [52,57]. The $E_p$ energy is relatively high in the Q&P steels with Si ≥ 1.5 wt.% and no well-defined unstable crack propagation stage is distinguished. The $P_M$ after tempering at 280 and 500 °C is higher than the observed maximum load of this steel after partitioning at 500 °C for 20 s.

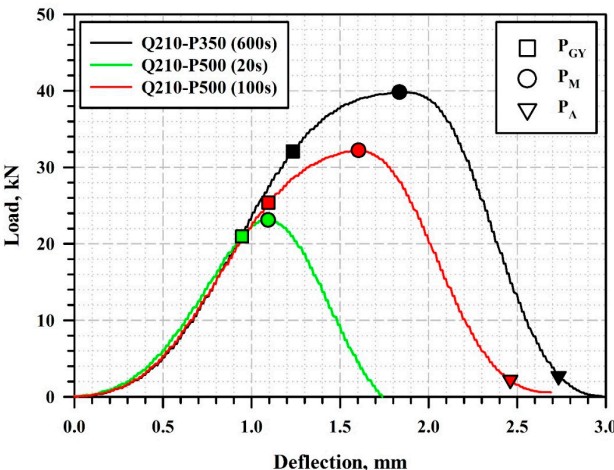

**Figure 8.** Load–deflection curves obtained during impact tests of the CVN specimens.

In order to analyze the dynamic behavior of the steel in different conditions, the stress dynamic yield strength (*DYS*) and dynamic ultimate tensile strength (DUTS) were calculated. The results are summarized in Table 7. The *DYS* represents the yielding criterion related to shear stress and was calculated using the following relation [54,55]:

$$DYS = \frac{3.732 P_{GY} W}{C_{GY}(W-a)^2 B} \tag{11}$$

where *W* = *B* = 10 mm are the specimen width and thickness, respectively, a = 2 mm is the notch depth and $C_{GY}$ is the constraint factor = 1.336 for standard ASTM tup. The *DUTS* was calculated using a relationship of the same form [4,11,54,55]:

$$DUTS = \frac{\eta_{Pm} P_m W}{(W-a)^2 B} \tag{12}$$

where $\eta_{Pm}$ = 2.531 is the empirical factor associated with the ratio between shear and tensile stress. It can be seen (Table 7) that the values of YS are slightly smaller than the *DYS* values for the steel subjected to partitioning at 350 °C for 600 s and 500 °C for 100 s. The crack nucleation is initiated after yielding and ductile crack growth occurs up to the $P_M$ point [52]. In contrast, YS > *DYS* in the steel after partitioning at 500 °C for 20 s and the crack initiation occurs due to plastic deformation at the notch-tip providing constraint condition [52,56]. Since *DYS* and *DUTS* are essentially the same for this steel condition, fast crack propagation occurs with a minor contribution from ductile fracture [4,11]. Fast crack propagation occurs under stress for brittle fracture that could be expressed by the following relationship [4,11,58,59]:

$$\sigma_F = 1.41 \sqrt{\frac{2E\gamma_s}{\pi(1-\nu^2)}} d_{eff}^{-1/2} \tag{13}$$

where *E* = 212 GPa is Young's modulus, $\gamma_s$ is the surface energy of the cleavage plane for the martensite structure, $\nu$ = 0.293 is Poisson's ratio and $d_{eff}$ is the effective grain size for the fracture. In contrast, the samples after partitioning at 350 °C for 600 s and 500 °C for 100 s exhibit DUTS significantly higher than YS in these conditions and fast crack propagation occurs with a high contribution from ductile fracture.

Inspection of Figure 8 shows that an increase in the CVN impact adsorbed energy is associated with increasing DYS and DUTS values and an expanded crack initiation stage. Q&P processing with partitioning at 350 °C for 600 s and 500 °C for 100 s provides ductile fracture behavior. Q&P processing with partitioning at 500 °C for 20 s leads to remarkable

embrittlement. For this sample, crack initiation occurs below the onset of yielding, and the fracture toughness is mainly controlled by $\sigma_F$ value [11].

### 3.4. Fractography

Figure 9 shows the overall views of the fracture surfaces after tensioning. All tensioned specimens show remarkable necking. An increase in partitioning temperature expands the fibrous zone (FZ). After partitioning at 350 °C for 600 s, crack initiation occurs in a mixed ductile–brittle manner (Figure 9(a2)) [60]. The main feature of the fracture surfaces is a small $d_{eff}$ for brittle fracture. Packets and some blocks play a role in the effective grain size [11]. Quasi-cleavage fracture takes place [4,37,38,53,54,60,61]. Areas of dimple fracture are separated by the cleavage units. Ductile fracture results in a dimple pattern in the shear-lip zone (SLZ) (Figure 9(a1)). Dimple dimensions decrease with increasing distance from the center to the edges of the tensioned specimen. After partitioning at 500 °C for 20 s, crack initiation in the FZ occurs through quasi-cleavage fracture (Figure 9(b2)), and crack propagation in SLZ takes place in a ductile manner with very fine dimples [60] (Figure 9(b1)). Intergranular fracture is observed in the FZ but its contribution to crack initiation is insignificant. After partitioning at 500 °C for 100 s, the contribution of intergranular fracture to crack initiation in the FZ increases while quasi-cleavage fracture plays the major role (Figure 9(c2)). Crack propagation in the SLZ occurs in a ductile manner (Figure 9(c1)). Weak evidence for decohesion [4,37,60] could also be found in the SLZ (Figure 9(c1)).

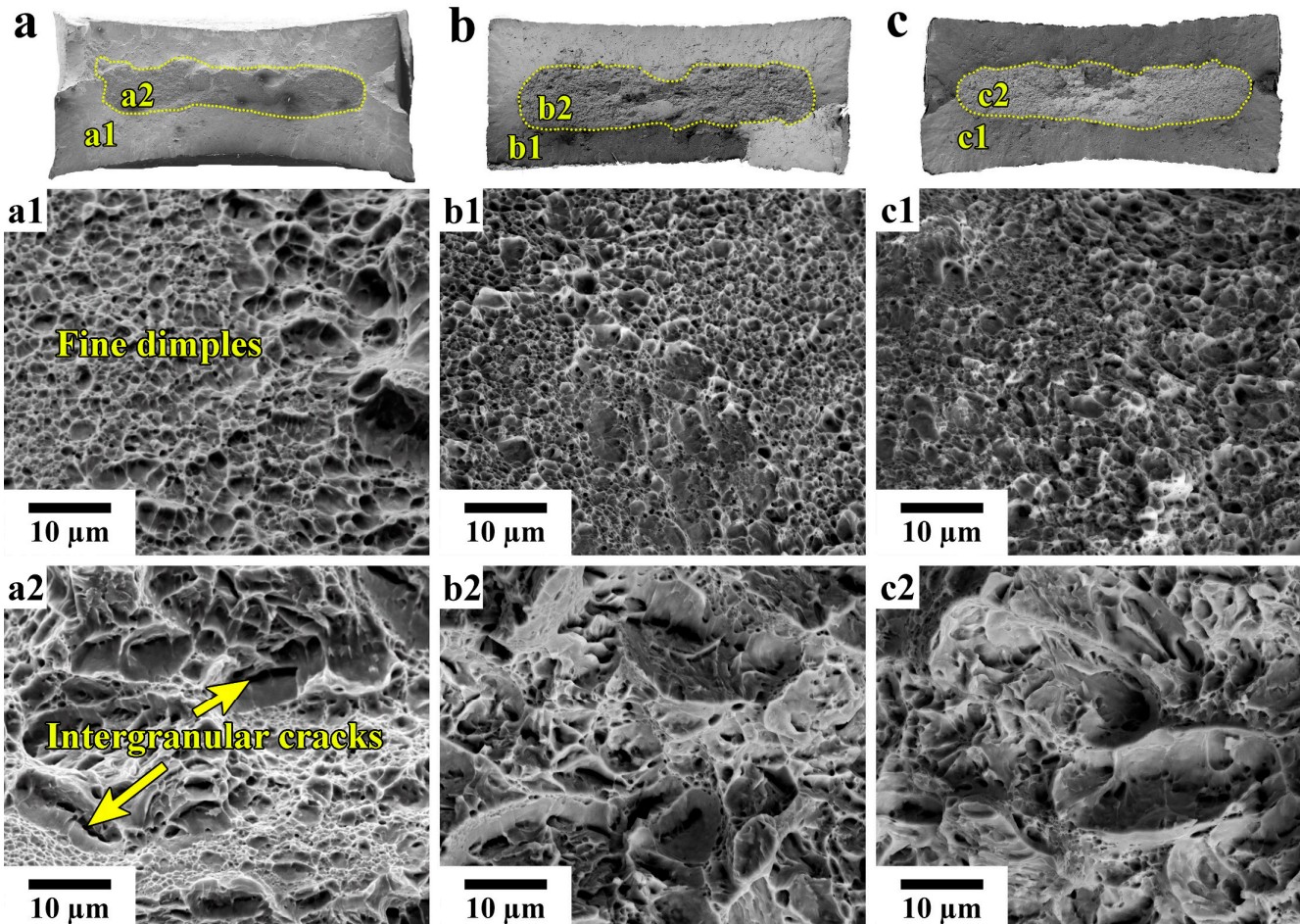

**Figure 9.** The overall view of fracture surfaces of tensioned specimens after partitioning at 350 °C for 600 s (**a**); partitioning at 500 °C for 20 s (**b**) and partitioning at 500 °C for 100 s (**c**). The enlarged SEM fractographs of the shear-lip zone (**a1–c1**) and fibrous zone (**a2–c2**).

The effect of Q&P processing on the area fractions of these zones is summarized in Table 8 and the fractographs of the V-notch Charpy specimens after impact tests are presented in Figure 10. The fracture surfaces of the CVN specimens consist of the initiation zone (IZ), fast crack propagation/fibrous zone (FCPZ) and shear-lip zone (SLZ) (Figure 10a–c) [37,57,61,62]. A dimple fracture is observed in the IZ for all steel conditions, indicating the occurrence of ductile fracture in this zone (Figure 10(a1–c1)). However, the parameters of the Q&P processing strongly affect the mechanisms of the crack propagation. In the steel after partitioning at 350 °C for 600 s, stable crack propagation leads to an increased size of dimples and mixed fracture takes place in the FCPZ (Figure 10(a2)). Coarse dimples alternate with areas of quasi-cleavage fracture. The observed cleavage coherent lengths are comparable with the size of martensite packets and block thickness. Therefore, the martensite packets and blocks play a role in the effective grain size for brittle fracture that provides small $d_{eff}$ and high $\sigma_F$ values [4,37,63]. In the latter case, crack propagation may occur in a ductile manner. The area fractions of dimples and quasi-cleavage are nearly the same. In addition, crack propagation occurs in a ductile manner at the lateral edges (Figure 10(a3)), which provides a high area fraction of SLZ (Table 8). F. Zia-Ebrahimi et al. [56] interpreted crack propagation in this zone in terms of the arrest of unstable crack propagation. However, the rectangular lateral surfaces (Figure 10a–c), the shape of the load–displacement curves (Figure 8) and relatively high $E_p$ values (Table 7) are indicative of the concurrent formation of FCPZ and SLZ at the lateral edges. The onset of crack arrest at small loads (Figure 8) and the very small area of the bottom SLZ (Figure 10a) support this conclusion.

**Table 8.** Effect of Q&P processing on the area fractions of initiation zone (IZ), fast crack propagation/fibrous zone (FCPZ) and shear-lip zone (SLZ).

| Temperature and Duration of Partitioning Stage | IZ, % | FCPZ, % | SLZ, % |
|---|---|---|---|
| 350 °C for 600 s | 4 | 71 | 25 |
| 500 °C for 20 s | 2 | 89 | 9 |
| 500 °C for 100 s | 2 | 81 | 17 |

The general view of the steel sample partitioned at 500 °C for 20 s (Figure 10b) suggests that crack initiation occurs in the center of the V-notch tip and subsequent crack propagation occurs in a ductile manner from the center of the V-notch tip to its edges. This fractography observation is consistent with the onset of ductile crack growth at the DYS < YS. The transition from ductile crack propagation in a transgranular manner to quasi-brittle crack propagation in a mixed transgranular and intergranular manner is observed in the FCPZ at a short distance from the V-notch flaw to the bottom of the CVN specimen (Figure 10(b2)). However, crack propagation at the lateral edges occurs in a fully ductile manner (Figure 10(b3)), which provides a remarkable $E_p$ value (Table 7) correlated with the significant fraction of SLZ (Table 8) located mainly at the lateral edges (Figure 10b). It is worth noting that the bottom SLZ has negligible width (Figure 10b) and is characterized by coarse and shallow dimples.

In the steel after partitioning at 500 °C for 100 s, stable crack propagation leads to the appearance of quasi-cleavage fracture areas. In the FCPZ, mixed quasi-cleavage and dimple fracture is observed (Figure 10(c2)). The coherent cleavage length is often restricted by the block boundaries. At the lateral edges, dimple fracture takes place (Figure 10(c3)). No difference between fracture surfaces at the bottom of the CVN specimen and in the FCPZ could be clearly distinguished. The portions of SLZ (Table 8) and degree of ductile fracture (Figure 10(c–c3)) increase with increasing duration of the partitioning stage at 500 °C.

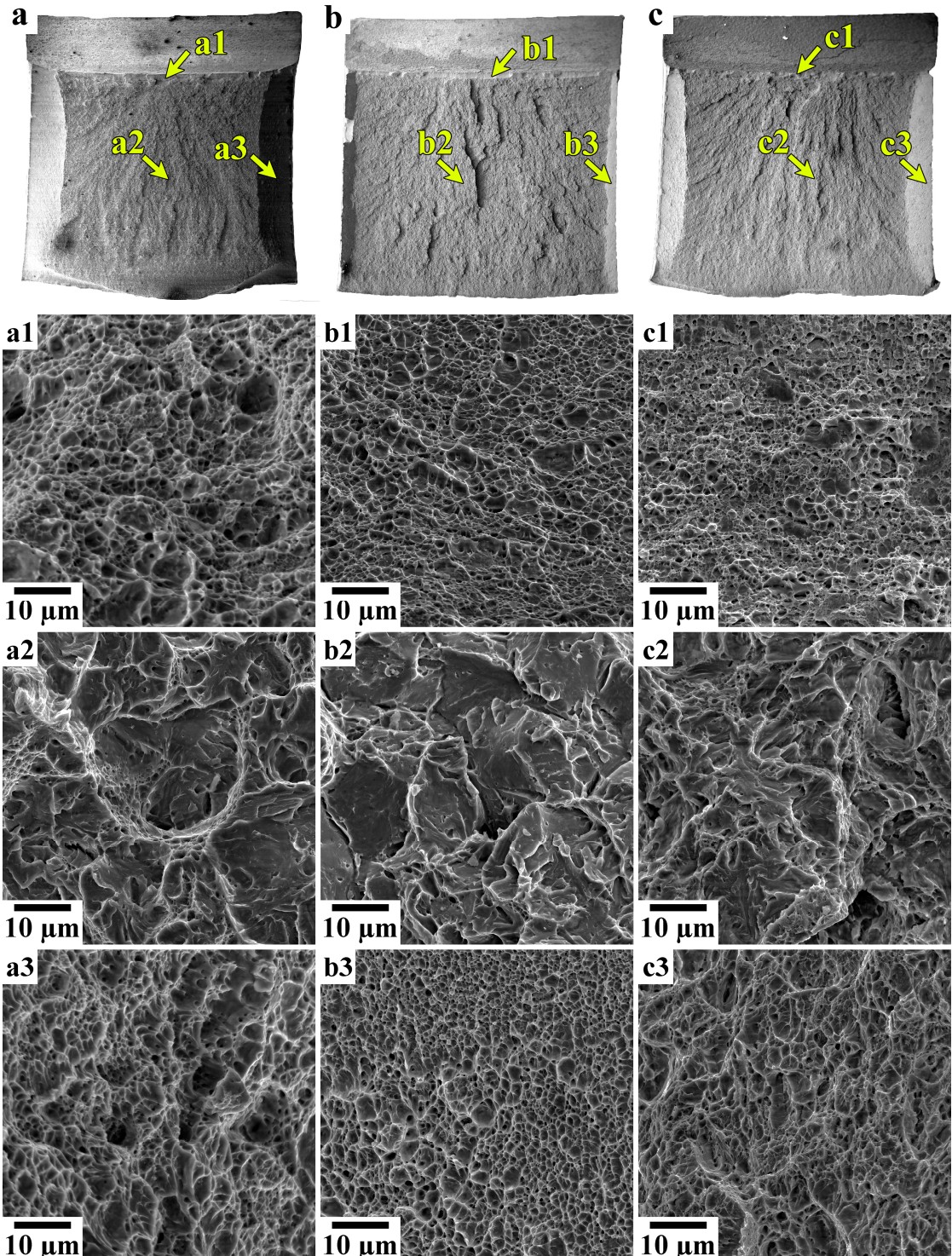

**Figure 10.** Low-magnification general view (**a–c**) and high-magnification fractographs of IZ (**a1–c1**), FCPZ (**a2–c2**) and SLZ at lateral edges (**a3–c3**) of the CVN specimens. Partitioning at 350 °C for 600 s (**a–a3**), partitioning at 500 °C for 20 s (**b–b3**) and partitioning at 500 °C for 100 s (**c–c3**).

## 4. Discussion

### 4.1. Effect of Q&P Processing on the Microstructure

The partitioning at temperatures of 350 and 500 °C leads to two important distinct features of the microstructure that strongly affect the mechanical properties and fracture

behavior. First, the volume fraction of the transition carbides in the primary martensite after partitioning at 350 °C is significantly higher than that after partitioning at 500 °C. These data support the presumption of J.D.Speer et al. [3,64] that the precipitation of transition carbides is facilitated at low temperatures, leading to strong carbon depletion from the martensitic matrix. At relatively high partitioning temperatures, the transition carbides precipitated at the quenching temperature tend to dissolve due to heating above the transition carbide solvus [3,65]. In addition, cementite precipitation takes place at $T \geq 474$ °C in this steel [4] and transition carbides dissolve under tempering/partitioning at 500 °C in accordance with the Gibbs–Thomson effect, providing a source of carbon for the formation of the more stable cementite [49]. The volume fraction of the transition carbides is small and the martensitic matrix remains carbon-enriched after partitioning at 500 °C for 20 s. After partitioning at 500 °C for 100 s, the volume fraction of cementite is negligible in comparison with conventional tempering at this temperature [4]. However, an increase in the partitioning time promotes the carbon depletion from the martensitic matrix for cementite formation.

Second, the carbon content in RA after partitioning at 350 °C is higher than that after partitioning at 500 °C and this difference is critical for the susceptibility of RA to transformation to strain-induced martensite under static and dynamic test conditions. J. D. Speer et al. showed [3,64] that carbon enrichment of RA during partitioning is strongly dependent on the volume fraction of martensite/ferrite. The volume fraction of martensite after quenching is similar for all partitioning conditions. However, the bainitic transformation at the partitioning stage at 350 °C is accompanied by an increase in the volume fraction of martensite/bainitic ferrite providing the enrichment of RA by carbon up to 1.3 wt.%. This RA is relatively stable against the transformation to secondary martensite under final cooling and strain-induced martensite under tension and impact tests. In contrast, the bainitic transformation is avoided during partitioning at 500 °C and a high portion of RA remains. The carbon enrichment is limited to ~1.0 wt.% and this RA is highly susceptible to transformation to the secondary martensite and strain-induced martensite.

*4.2. Structure and Mechanical Properties of Q&P Steels*

Inspection of the experimental results shows that the relationship between key mechanical properties such as strength, ductility and fracture toughness is very complicated. In contrast with other types of steels [38], the studied Q&P steel could be strong and tough, concurrently, after partitioning at 350 °C and both highly ductile and highly brittle after partitioning at 500 °C for 20 s. The unusual relationships between strength, ductility and fracture toughness are associated with the complex microstructure that evolved after Q&P processing and the fact that different structural constituents control different mechanical properties. The highest YS is attained if Q&P processing provides the martensitic matrix with a large amount of transition carbides and film-like RA with nanoscale thickness and enriched by carbon. The YS of the martensitic matrix [54] and the YS of RA [11] have to be essentially the same. After partitioning at 350 °C, this combination of structural constituents evolved and the Q&P steel exhibits an unusual combination of a YS of 1310 MPa and a CVN impact energy of 58 J.

YS is mainly controlled by the strength of the martensitic matrix. Dense precipitation of transition carbides provides the highest strength of primary martensite [4,37,53]. Concurrently, the depletion of carbon from the martensitic matrix highly increases the $\gamma_s$ value in Equation (13) and, therefore, the fracture stress, $\sigma_F$ [37,65]. A three-fold increase in the CVN impact energy of the present steel subjected to partitioning at 350 °C in comparison with this steel tempered at 280 °C [4] could be attributed to the higher degree of carbon depletion from the martensitic matrix to RA, which increases the $\gamma_s$ value in Equation (13). In addition, the volume fraction of RA after Q&P is more than 10 times higher than that in the quenched and tempered steel [4]. Film-like RA is located at a major portion of the block boundaries and these boundaries act as obstacles to cleavage propagation [37]. A strong refinement of the effective grain size for the fracture, $d_{eff}$, takes place due to the

decrease in the coherent cleavage length from the $d_{eff} \approx D_{packet}$ in tempered steel [4] to the $d_{eff} \approx D_{block}$ in the steel after partitioning at 350 °C. As a result, DYS becomes higher than YS and ductile fracture provides a significant contribution to the fracture process. It is worth noting that the observed increase in the elongation-to-failure and PSE in comparison with this steel tempered at 280 °C [4] could be attributed to the transformation of blocky-type RA to strain-induced martensite. The observed increment in ductility and the PSE of ~60% in comparison with the tempered steel [4] is attributed to the extension of the uniform elongation stage.

The transformation of RA to strain-induced martensite under tension in the steel after partitioning at 500 °C unambiguously showed that film-like RA is not susceptible to martensite transformation if carbon enrichment provides a low $M_s$ temperature. This is an important complement to the findings of a previous study [30]. The steel sample after partitioning at 500 °C for 20 s exhibits the transformation-induced plasticity (TRIP) effect, which increases both the strength and ductility [16,66]. The formation of strain-induced martensite raises the strain-hardening rate under tension, which highly increases the uniform elongation since a high value of $d\sigma/d\varepsilon$ leads to a sharp increase in the flow stress within the necked region, providing resistance to further strain localization. As a result, a high elongation-to-failure is achieved. Concurrently, a high UTS is attained due to extensive strain-hardening despite a relatively low YS. The "composite" law based on the linear additive strengthening contributions is commonly used to estimate the YS of the steels with multiphase microstructures [11]:

$$YS = \sum_{i=1}^{n} \sigma_{0.2}^i f^i \tag{14}$$

Thus, the decreased YS in the steel after partitioning at 500 °C can be attributed both to the low precipitation hardening in the primary martensite due to a decreased fraction of transition carbides and to the low strength of RA due to the relatively small content of carbon, which is an effective element for solid solution strengthening [11]. Nevertheless, high values of the PSE could be attained in Q&P steel if the TRIP effect takes place. It is worth noting that the fracture mechanisms are similar in all tensioned specimens, and thus, have no significant effect on the elongation-to-failure.

It is obvious that the fracture toughness is controlled by the carbon content of the martensitic matrix. After partitioning at 500 °C for 20 s, the depletion of carbon from martensite is insignificant in comparison with that after tempering at this temperature and the CVN impact energy is smaller by a factor of ~2. A low fracture toughness is observed despite the fact that YS is not high. Increasing the duration of the partitioning stage promotes the depletion of carbon from the martensite, which increases the fracture toughness. The CVN impact energy becomes even higher than that after conventional tempering at this temperature due to the smaller effective grain size for fracture [37]. Thus, no correlation between strength, ductility and fracture toughness is observed in Q&P steels if these materials have distinct structural components.

### 5. Conclusions

A 0.33% C-1.85% Si-1.44% Mn-0.58 % Cr steel was processed by a two-step Q&P treatment including quenching at 210 °C for 30 s followed by partitioning at 350 °C and 500 °C and quenching to room temperature.

1.  Partitioning at a temperature of 350 °C for 600 s produces a martensitic matrix with a large amount of η–$Fe_2C$ carbides, carbide-free bainitic ferrite and retained austenite with a carbon content of ~1.3 wt.%. The volume fraction of secondary martensite in the steel in this condition is negligible. A major portion of retained austenite exhibits a film-like shape and is unsusceptible to transformation to strain-induced martensite. The steel in this condition is strong (yield strength is 1310 MPa) and tough (the CVN impact energy is 58 J), but ductility (10%) and the product of strength and elongation (15.6 GPa·%) are not high.

2. Partitioning at a temperature of 500 °C for 20 s produces a martensitic matrix with a low density of η–Fe$_2$C carbides, secondary untempered martensite and retained austenite with a carbon content of ~1.0 wt.%. This retained austenite is susceptible to partial transformation to strain-induced martensite during tensile testing. The steel in this condition is highly ductile with an elongation-to-failure of 18% and the product of strength and elongation of 25.8 GPa·%, but brittle (the CVN impact energy is 18 J) despite a relatively low yield strength (1180 MPa).

3. Increasing the duration of the partitioning stage at 500 °C up to 100 s leads to the precipitation of cementite, which decreases the yield strength (950 MPa), ductility (14%) and product of strength and elongation (20.8 GPa·%) but increases the fracture toughness (the CVN impact energy is 38 J).

4. There is no correlation between strength, ductility and fracture toughness in Q&P steels if the processing produces structures distinct in their structural components. The martensitic matrix and a dispersion of carbides control the strength and fracture toughness, while ductility and the product of strength and elongation are dependent on the volume fraction and morphology of the retained austenite and its susceptibility to transformation to strain-induced martensite.

5. An enhanced balance between strength, ductility and impact toughness in 0.33 wt.% C Q&P steel can be achieved by forming a microstructure containing a significant fraction of RA with optimal stability and a low fraction of the brittle secondary martensite. This balance could be tailored by increasing the partitioning temperature to avoid the bainitic transformation and by controlling the isothermal holding time at the partitioning stage to reduce the fraction of secondary martensite.

**Author Contributions:** Conceptualization, E.T. and R.K.; methodology, E.T., S.B. and Y.B.; validation, Y.B. and E.T.; formal analysis, S.B. and Y.B.; investigation, S.B., E.T., Y.B., T.K. and R.K.; resources, R.K.; data curation, E.T., T.K. and S.B.; writing—original draft preparation, E.T. and R.K.; writing—review and editing, Y.B., E.T. and S.B.; visualization, E.T.; supervision, R.K.; project administration, R.K.; funding acquisition, R.K. All authors have read and agreed to the published version of the manuscript.

**Funding:** This research was funded by the Ministry of Science and Higher Education of the Russian Federation, grant number 075-15-2021-572.

**Data Availability Statement:** Not applicable.

**Acknowledgments:** The studies were carried out on the equipment of the Joint Scientific Center for Technologies and Materials of Belgorod State National Research University, which was supported by the Ministry of Science and Higher Education of the Russian Federation under contract No 075-15-2021-690 (unique identifier RF----2296.61321X0030).

**Conflicts of Interest:** The authors declare no conflict of interest.

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
