# Peer review of "Relationships between Strength, Ductility and Fracture Toughness in a 0.33C Steel after Quenching and Partitioning (Q&P) Treatment"

_crystals, doi:10.3390/cryst13101431_

Round 1

Reviewer 1 Report

This study investigated the influence of quench and partitioning (Q&P) treatment on the mechanical behaviors of steels. The manuscript is well written and easy to read. In-depth microstructure characterization was performed by SEM, EBSD, and TEM. Mechanical properties were examined using tension and impact tests. Nonetheless, several issues need to be addressed before the manuscript can be accepted for publication.

1. The authors referred Fig. 4 (a-c) as misorientation map, however, this term is improper and these maps should be invers pole figures or grain orientation maps. Widely used misorientation maps such as Kernal Average Misorientation (KAM) maps is different from the figures shown here. For Fig. 4(a1-c1) what is the color code for these figures? If the authors would like to highlight the HAGBs, it’s better to use bright colors like red to label the boundaries.

2. Have the authors checked the repeatability of the tensile and impact test results? Only one curve for each specimen were shown in Fig. 7 and Fig. 8.

3. In Fig. 7(b), the authors can consider drawing the strain hardening rate and true stress as a function of true strain in the same graph, so that the intersection between the strain hardening and true stress-strain curves can be better determined (especially by the readers).

4. This study is systematic but reported very limited new insights to the metallurgy community. Although the authors already stated in the manuscript that no straightforward correlation can be established between the processing methods and the mechanical performance of the Q&P steels, can the authors comment on possible routes to improve the mechanical properties of steels based on their study?

Author Response

Dear Reviewer,

Thank you for your positive evaluation of our manuscript and useful remarks to improve the paper. We have revised the paper according to the recommendations as described below. The respective changes have been highlighted in the revised version.

  1. The authors referred Fig. 4 (a-c) as misorientation map, however, this term is improper and these maps should be invers pole figures or grain orientation maps. Widely used misorientation maps such as Kernal Average Misorientation (KAM) maps is different from the figures shown here.

A1. The caption of figure 4 was corrected in accordance with the Reviewer’s recommendation.

For Fig. 4(a1-c1) what is the color code for these figures? If the authors would like to highlight the HAGBs, it’s better to use bright colors like red to label the boundaries.

A1.1.The color code for Fig. 4(a1-c1) was introduced in the revised version of the manuscript and the grain boundaries were described in the figure caption as follows:

“…EBSD maps showing the identification numbers of martensite packets and the reconstructed boundaries of PAGs (thick black lines) and HABs (thin black lines) …”

  1. Have the authors checked the repeatability of the tensile and impact test results? Only one curve for each specimen were shown in Fig. 7 and Fig. 8.

A2. The reliability of the tensile properties was ensured by testing two samples for each material condition. The description of the tensile tests in the Material and methods section was amended as follows:

“…Two tensile specimens with a gauge length of 35 mm and a cross-section of 7mm x 3mm were tested for each material condition.”

The observed scatter in the tensile properties is given in Table 6 of the revised manuscript.

The repeatability of the impact toughness values was ensured by testing two CVN specimens for the material condition with the highest adsorbed energy. The obtained impact toughness values for this material condition were 59 J and 56 J. For the subsequent analysis the average value of the CVN toughness of 58J and the curve for 56 J were used.

  1. In Fig. 7(b), the authors can consider drawing the strain hardening rate and true stress as a function of true strain in the same graph, so that the intersection between the strain hardening and true stress-strain curves can be better determined (especially by the readers).

A3. The Fig. 7b was amended in the revised manuscript in accordance with the Reviewer’s recommendation.

  1. This study is systematic but reported very limited new insights to the metallurgy community. Although the authors already stated in the manuscript that no straightforward correlation can be established between the processing methods and the mechanical performance of the Q&P steels, can the authors comment on possible routes to improve the mechanical properties of steels based on their study?

A4. The conclusion section was expanded by addition of the statement based on the revealed correlation between the microstructure and mechanical properties to provide the useful information for the metallurgy community:

“…5. The enhanced balance between strength, ductility and impact toughness in 0.33wt.% C Q&P steel can be achieved by forming the microstructure containing significant fraction of RA with optimal stability and low fraction of the brittle secondary martensite. This balance could be tailored by increasing the partitioning temperature to avoid the bainitic transformation and by controlling the isothermal holding time at the partitioning stage to reduce the fraction of secondary martensite.

Reviewer 2 Report

The mnuscript studies relationships between strength, ductility and fracture toughness in a 0.33C steel after quenching and partitioning (Q&P) treatment. The manuscript needs a significant imporvement:

1- The introduction is not complete and do not show the motivation of the work. 

2- If there is no correlation between strength, ductility and fracture toughness is observed in Q&P steels if these materials are distinctly distinguished by structural constituents, it seems the motivation of the current work is not enough. Please explain in detail and change the story of the manuscript. I think there are some interesting findings however, some of them are not directly related to the relationships between strength, ductility and fracture. 

3- Regarding the orientation relationship between the austenite and martensite and the other phases, please expand the introduction and explain the orientations in detail by citing the following references:

https://doi.org/10.1016/j.jmrt.2022.12.016

https://doi.org/10.1016/j.ijhydene.2023.03.396

4- The fracture surfaces in Fig. 9 are not clear. Please magnify the figure and explain more about the fracture surface and the indications.

5- The authors used the composite Hall-Petch relationship for other alloys, is that also possible to use that for the current study? Please add comments about that.

It is clear and understandable.

Author Response

Dear Reviewer,

Thank you very much for sending us the useful comments regarding our paper. We have revised the paper according to the recommendations as described below. The respective changes have been highlighted in the revised version.

1- The introduction is not complete and do not show the motivation of the work. 

A1- To clarify the motivation of the present study we revised the introduction part and highlight the importance of the work as follows:

Unfortunately, the effect of the precipitation of carbide particles and characteristics of RA on the combinations of strength, ductility and impact toughness is scarcely reported for the Q&P steels.

The aim of the present work is to examine the relationships between strength, ductility, PSE, and CVN impact absorbed energy in a low-alloy steel with 0.33wt.%C processed by Q&P. The effects of the tempering of the primary martensite and the stability of RA on the mechanical properties are discussed. For this purpose, three different Q&P routes were performed to obtain the microstructures containing various volume fractions of bainitic ferrite and secondary martensite, as well as the different morphology and carbon content of RA…

2- If there is no correlation between strength, ductility and fracture toughness is observed in Q&P steels if these materials are distinctly distinguished by structural constituents, it seems the motivation of the current work is not enough. Please explain in detail and change the story of the manuscript. I think there are some interesting findings however, some of them are not directly related to the relationships between strength, ductility and fracture. 

A2- In addition to the expanded introduction part we modify the conclusions to highlight the revealed correlation between the microstructure and mechanical properties:

  1. The enhanced balance between strength, ductility and impact toughness in 0.33wt.% C Q&P steel can be achieved by forming the microstructure containing significant fraction of RA with optimal stability and low fraction of brittle secondary martensite. This balance could be tailored by increasing the partitioning temperature to avoid the bainitic transformation and by controlling the isothermal holding time at the partitioning stage to reduce the fraction of secondary martensite.

3- Regarding the orientation relationship between the austenite and martensite and the other phases, please expand the introduction and explain the orientations in detail by citing the following references:

https://doi.org/10.1016/j.jmrt.2022.12.016

https://doi.org/10.1016/j.ijhydene.2023.03.396

A3- The discussion on the relationship between the austenite and martensite was introduced in the introduction part of the present study with the help of the mentioned references as follows:

“…For low- and medium-carbon steels the both primary and secondary martensite usually holds the Kurdjumov–Sachs (K-S) orientation relationship with the austenite matrix that can be determined by the electron backscatter diffraction (EBSD) analysis [5,11-13]…”

4- The fracture surfaces in Fig. 9 are not clear. Please magnify the figure and explain more about the fracture surface and the indications.

A4- The fractographs in fig. 9 were enlarged and the characteristic features of the fracture were indicated on this figure in the revised manuscript.

5- The authors used the composite Hall-Petch relationship for other alloys, is that also possible to use that for the current study? Please add comments about that.

A5- The “composite” law based on the linear additive strengthening contributions was introduced in the discussion part of the revised manuscript to analyze the change in the yield strength of the studied steel as follows:

“… Concurrently, a high UTS is attained due to extensive strain-hardening despite a relatively low YS. The “composite” law based on the linear additive strengthening contributions is commonly used to estimate the YS of the steels with multiphase microstructures [12]:

                                                                        (14)

Thus, the decreased YS in the steel after partitioning at 500°C can be attributed both to the low precipitation hardening in the primary martensite due to decreased fraction of transition carbides and to the low strength of RA due to the relatively small content of carbon which is an effective element for the solid solution strengthening [12].”

The manuscript has been revised carefully to enhance the quality of English language.

Round 2

Reviewer 1 Report

The authors have adequately addressed my previous comments. The manuscript can be accepted for publication.

Reviewer 2 Report

The answers to the comments and the corrections are satisfactory and therefore, I recommend publishing the manuscript.

The English language and style of the manuscript require a minor revision.